# Music and Psychology & Social Connections Program: Protocol for a Novel Intervention for Dyads Affected by Younger-Onset Dementia

**DOI:** 10.3390/brainsci12040503

**Published:** 2022-04-15

**Authors:** Samantha M. Loi, Libby Flynn, Claire Cadwallader, Phoebe Stretton-Smith, Christina Bryant, Felicity A. Baker

**Affiliations:** 1Department of Psychiatry, The University of Melbourne, Parkville, VIC 3052, Australia; 2Neuropsychiatry, John Cade Level 2, Royal Melbourne Hospital, Parkville, VIC 3050, Australia; 3Department of Fine Arts and Music, The University of Melbourne, Parkville, VIC 3052, Australia; libby.flynn@unimelb.edu.au (L.F.); claire.cadwallader@unimelb.edu.au (C.C.); phoebe.stretton@unimelb.edu.au (P.S.-S.);felicity.baker@unimelb.edu.au (F.A.B.); 4Turner Institute for Brain and Mental Health, School of Psychological Sciences, Monash University, Clayton, VIC 3800, Australia; 5Department of Psychology, The University of Melbourne, Parkville, VIC 3052, Australia; cbryant@unimelb.edu.au; 6Music Education and Music Therapy Department, Norwegian Academy of Music, 0369 Oslo, Norway

**Keywords:** music-based therapies and intervention, dementia, therapeutic songwriting, younger-onset dementia, psychotherapy, online

## Abstract

Psychosocial interventions targeting the specific needs of people affected by younger-onset dementia are lacking. Younger-onset dementia refers to dementia where symptom onset occurs at less than 65 years old. Because of its occurrence in middle age, the impact on spouses is particularly marked and dyadic-based interventions are recommended. Music And Psychology & Social Connections (MAPS) is a novel online intervention, informed by the theory of adaptive coping by Bannon et al. (2021) for dyads affected by younger-onset dementia. MAPS combines therapeutic songwriting, cognitive behaviour therapy, and a private social networking group that focuses on the dyads. This will be a randomised controlled trial with a waitlist control. The primary aims are to assess whether MAPS improves depressive, anxiety, and stress symptoms in caregivers, with secondary aims to assess whether MAPS improves depressive symptoms in people with younger-onset dementia. The trial also aims to assess dyadic social connectedness; caregiver coping skills; and neuropsychiatric symptoms in people with younger-onset dementia. We will recruit 60 dyads to participate in a group-based weekly online program for 8 weeks facilitated by a credentialed music therapist and psychologist. Sessions 1 and 8 will include both caregivers and people with younger-onset dementia and Sessions 2–7 will involve separate group sessions for caregivers and those with dementia. There will be focus groups for qualitative feedback. Due to its online administration, MAPS has the potential to reach many dyads affected by younger-onset dementia.

## 1. Introduction

Dementia is a progressive neurodegenerative condition that affects cognition and many areas of functioning. Younger-onset dementia (YOD) is characterised by symptom onset before the age of 65 years [1]. YOD is associated with unique challenges and significant psychosocial impacts for the individuals living with YOD and their families, due to its onset at an earlier life stage when people are often gainfully employed, providing for their family, and managing multiple roles [2]. Specific challenges include heterogeneity of symptom onset [3,4], delay to and difficulties obtaining diagnosis [5], and problems accessing appropriate services [6]. Post-diagnostic services are often geared towards older adults with dementia and their caregivers [6,7], resulting in younger caregivers feeling isolated, vulnerable, less supported, and at higher risk of adverse mental health [8].

Receiving a diagnosis of dementia during middle age can be devastating, with individuals reporting feelings of powerlessness, hopelessness, loss of self-identity, and social exclusion [9,10,11]. Changes in the spousal relationship, a need for “normalcy” and peer support have also been endorsed as losses [10,12]. In addition, many people living with YOD commonly experience behavioural and personality changes. These neuropsychiatric symptoms include agitation, depression, apathy, and social withdrawal [13]. Collectively, these symptoms contribute to poor quality of life (QOL) in those with YOD [14], adverse effects in caregivers [15], and can precipitate early entry to residential care [16].

The majority of informal support and care is provided by partners of people living with YOD [7]. Younger caregivers report higher levels of burden and caring challenges compared to caregivers who are of older age and support people with older-onset dementia [17,18]. While there is some overlap between the experiences of younger and older caregivers, such as feelings of grief and loss, younger caregivers face particular challenges in relation to role changes, difficulties in planning for the future [19], and feeling more socially isolated [6]. The impact of YOD on the functioning and well-being of the family unit, especially the partner dyads, suggests interventions that target families/dyads may be more beneficial than those that focus solely on caregivers or people living with YOD individually [10,12,20].

For dyads living with dementia, several psychosocial intervention studies have demonstrated positive outcomes for both the caregivers and the people living with dementia. Dyadic interventions described in the research include a combination of psychoeducation, cognitive-behavioural therapy (CBT), and strategies for managing cognitive deficits and neuropsychiatric symptoms in dementia. The Acquiring New Skills While Enhancing Remaining Strengths program (ANSWERS) is a six-week strengths-based program comprising modules covering psychoeducation of dementia, communication skills, managing memory and strategies, physical activity, and recognising emotions and behaviours (Judge et al., 2010). The controlled trial reported that caregivers in the intervention group had improved mental health with reduced depressive and anxiety symptoms, less dyadic strain and role captivity, and improved mastery compared to controls [21]; however, outcomes for the individuals living with dementia have not been published. This face-to-face one-on-dyad intervention reported a high attrition rate (24%) in the intervention group, compared to 2% in the control group, with the authors hypothesising that the intervention may have been too time-consuming and overwhelming for the dyads [21].

The Danish Alzheimer Intervention Study (DAISY) was a randomised controlled multi-component and semi-individualised intervention for people 12 months post-diagnosis of Alzheimer’s disease (AD) and their caregivers [22]. In contrast to ANSWERS, DAISY employed a combination of group, one-on-dyad, and individual sessions with caregivers and individuals living with dementia alone. Furthermore, DAISY was an intervention of longer duration compared to ANSWERS—lasting between 8 to 12 months and included counselling and psychoeducation administered by a trained nurse and experts in dementia care, with individual follow-up telephone counselling. Separate group sessions for the people living with dementia and the caregivers occurred, whereby topics such as causes, treatment, legal issues, and social support were discussed. Of the dyads randomised to DAISY, there was an improvement in depressive symptoms in the people living with dementia but no differences were found in caregivers’ depressive symptoms or quality of life [22]. Although this intervention had multiple components and was of longer duration (12 months) compared to ANSWERS (6 weeks), it is unclear whether the length of intervention and which elements of an intervention (for example group vs. individual, psychoeducation vs. counselling) are more effective in improving depression and anxiety in people living with dementia and their caregivers.

While traditionally psychosocial interventions have been delivered face-to-face, online administration may have advantages in terms of flexibility and less time commitment due to minimal requirement for transportation [23]. These aspects might be particularly appealing for people living with YOD and their caregivers [6]. In addition, because YOD represents approximately 5–10% of all dementias [24,25], an online intervention would have the potential to reach many more YOD dyads, with a better cost-benefit compared to face-to-face administration [26]. In particular, with the impact of the COVD-19 pandemic, particularly on vulnerable populations, the capacity to provide interventions online has become a crucial requirement; however, research in this specific area remains limited. The Research to Assess Policies and Strategies for Dementia in the Young (RHAPSODY) pilot study by Metcalfe et al. [26] was a 6-week online psychoeducation about YOD and a skill-building program, which could be accessed freely by the caregivers involved. Content included the diagnosis of YOD, medical explanations, common problems and solutions, management of cognitive and behavioural symptoms, adapting to relationship changes, support, and self-care, using written information, videos and case studies. Participants accessed it a mean of 7.5 times over the 6 weeks and viewed 31% of the online content. There were no differences in caregiver burden and quality of life between participants and waitlist controls. Feedback from participants focused on the lack of access to a specific person to discuss aspects of the content of the program. Having an online facilitator to guide sessions and set goals may lead to better acceptability and feasibility, as shown in the Partner in Balance program, another online intervention for caregivers of people living with YOD [27]. This program included case studies and written material, but no specific information about YOD itself; however, additional content covered areas such as combining care with work, sexuality/intimacy, communication skills, acceptance, focusing on the positives, self-understanding, insecurities, and rumination. Modules that acknowledged the losses but also encouraged some positives were reported to be particularly helpful for caregivers.

Looking to address the social isolation reported by both caregivers and people living with YOD, group programs may provide peer support and the feeling that “one is not alone” [28]. Therapeutic songwriting is gaining recognition as an effective group music therapy intervention for people living with dementia, their caregivers, and the dyad. Therapeutic songwriting embeds various psychological approaches and can engage and motivate people, foster experiences of mastery, develop participant self-confidence, enhance self-esteem, develop or transform a sense of self, assist with insight and clarifying, externalising thoughts and emotions, and be a tool to tell participants’ stories [29,30]. Songwriting is an effective medium to explore personal issues because it is versatile, involves a powerful combination of language and music, is a culturally appropriate form of self-expression, can be social, and invites collaboration and a therapeutic relationship [31,32]. There is growing research on group therapeutic songwriting with people living with dementia, and caregivers of people with dementia [30,31,33] participating in separate groups. Recent studies have also explored group therapeutic songwriting with people living with dementia and caregiver dyads [29,34]. Participants across these studies reported the songwriting experiences as stimulating, collaborative and enjoyable. Among dyads that consisted of older adults living with dementia and their caregivers, therapeutic songwriting was found to be an acceptable and feasible intervention, with a trend toward decreased depression in the people living with dementia [29]. Therapeutic songwriting, however, has not yet focused on people with YOD and their caregivers. This approach may be favoured as a psychotherapeutic tool for people with YOD and their caregivers as non-threatening and safe, traversing cultural and language barriers with the potential for good participation within a group context [35] and has previously been shown to be feasible in an online context [36]. In addition, online engagement and social networking have been positively linked to improved opportunities for connection and interaction by providing a platform for sharing experiences which may also be appealing to the younger age group [37].

As far as we are aware, there have been no published interventions that focus on the dyad in YOD. Given the need to design psychosocial interventions that address the dyad as a unit, are accessible for busy caregivers, and are engaging and calming for people living with YOD, a combined music therapy and psychotherapy program delivered online, Music And Psychology & Social Connections (MAPS) was designed and will be tested using a pragmatic, wait-list control trial design. The present study aims to improve the mental health and social connectedness in caregivers and people with YOD, improve coping skills in caregivers, and decrease neuropsychiatric symptoms in people with YOD. The primary hypothesis is that the MAPS program will have a larger effect on improving depressive, anxiety, and stress symptoms in caregivers compared with standard care. Secondarily, we hypothesise that compared with standard care, MAPS will lead to greater improvements in: (a) depressive symptoms in people living with YOD; (b) the effectiveness and types of coping strategies utilised by caregivers; (c) levels of social connectedness in people living with YOD and their caregivers; and (d) the levels of neuropsychiatric symptoms in the people living with YOD.

## 2. Materials and Methods

### 2.1. Study Design

This pragmatic randomised waitlist control trial design with post-program focus groups will test the primary and secondary hypotheses with a 1:1 allocation ratio. People living with YOD and caregiver dyads will be randomised to one of two conditions: (1) MAPS; or (2) Waitlist control (Figure 1). Dyads randomised into the MAPS group will receive 8 × 100-min group sessions of combined psychotherapy and music therapy, supplemented with an optional online closed Facebook group with activities to be completed between sessions. Dyads will attend sessions 1 and 8 online together; however, from weeks 2 to 7, caregivers and people living with YOD will participate separately in online groups. Participants in the standard care group will participate in the MAPS program following completion of the post-measures. Data will be collected at baseline and the end of the 8-week intervention period. This trial is framed as a superiority trial where we hypothesise that the MAPS group will be superior to usual care for the primary outcome measure.

### 2.2. Intervention Description

#### 2.2.1. The Conceptual Framework

The conceptual framework (Figure 1) underpinning the combined music therapy and psychotherapy intervention (MAPS) is built around enabling dyads to address the issues of loss and changes for caregivers and people living with YOD [10,12]. Bannon et al. [12] for example, described seven adaptive coping strategies. These include: (1) increasing the couple’s ability to process, normalise and accept negative emotions; (2) promoting normalcy; (3) preserving the independence of the person living with YOD; (4) developing ways to improve collaborative communication; (5) maintain existing and develop new social support networks; (6) introducing lifestyle changes and strategies for self-care including mindfulness and recreational pursuits; and (7) meaning-making, humour, and positivity. In our framework, focused group discussions enable people living with YOD and their caregivers to explore each of these coping strategies in a safe and accepting environment. The sessions will be manualised so the program can reliably be replicated and administered. To further reinforce interpersonal learning, insight, validation of experiences, and connection with others, the groups then engage in a therapeutic songwriting experience. For those people living with YOD who may experience challenges with language expression, music and songwriting may assist in expressing feelings in ways they cannot find words for.

#### 2.2.2. Description of the MAPS Program

MAPS (registered with the Australian and New Zealand Register of Clinical trials, ACTRN12622000326796p) is an 8-week online program for people living with YOD and their caregivers that embeds the adaptive coping model by Bannon et al. (2021), using a combination of therapeutic songwriting, cognitive-behavioural therapy, and social media participation (Figure 2).

Private Facebook groups will provide space for group members to interact between sessions and develop adaptive coping and build connections through music and social media. The MAPS program addresses one adaptive coping topic each week (Figure 1). A music therapist and psychologist will co-facilitate sessions with the caregivers and people living with YOD together (weeks 1 and 8) and separately (weeks 2–7). Sessions will include group discussions using cognitive-behavioural therapy frameworks on the weekly topic followed by group therapeutic songwriting experiences where the group crafts lyrics and music that reinforce the key discussion points. Homework tasks supported by the online Facebook group will encourage dyads to further discuss and explore the topics raised in weekly sessions, which will help to further reinforce learnings and provide opportunities for relationship building and adaptive coping strategies. Dyads will be encouraged to share music they enjoy. The final session (week 8) will bring together the dyads to explore meaning-making, humour, and positivity and provide an opportunity to share the group composed songs for closure. It is hoped that playing the songs composed during the program to the group will provide a positive tangible sense of achievement and mastery. The songs produced by MAPS will be audio-recorded so that these can be played during presentations to demonstrate what can be achieved. Caregivers and people with YOD will receive dedicated workbooks (emailed or posted depending on preference) which will be used during the 8 weeks. The online platform Zoom will be used.

#### 2.2.3. Training and Assessment of Fidelity

A careful plan for the fidelity of the study design, treatment integrity, treatment differentiation, treatment receipt, and treatment enactment has been developed. A manual was developed prior to implementation as well as a fidelity checklist to be completed by all therapists at the end of each session. Sessions will be audio-recorded to monitor fidelity and a session will be randomly selected for fidelity assessment by a member of the research team who is not a facilitator. Individualised supervision and monitoring of therapists will be conducted fortnightly to address any challenges associated with program implementation according to the manual.

### 2.3. Participants and Recruitment

#### 2.3.1. Inclusion and Exclusion Criteria

Sixty dyads (sixty caregivers, sixty people living with YOD) living in Australia will be recruited. There is no restriction on the type of YOD. Participants will be recruited by advertising the study at the Royal Melbourne Hospital (such as the Neuropsychiatry YOD clinic), the Step-Up for Dementia Research Database, on social media, and through other forums such as Dementia Australia. Caregivers will be eligible to participate if they: (1) self-identify as a caregiver of a person living with YOD; (2) consider themselves in a long-term relationship with the person living with YOD (mostly will be spouses, but are not restricted); (3) score either ≥14 on the Depression subscale of the Depression Anxiety and Stress Scale, DASS-21 [38], ≥12 on the Anxiety subscale of the DASS, or ≥20 on the Stress subscale; (4) able to understand and communicate in English; (5) have functional hearing (with or without hearing aids or other devices); (6) have access to the internet and a device (laptop or tablet) at home; and (7) have an existing Facebook account or be willing to create one. Participants living with YOD will be eligible if they: (1) have a diagnosis of YOD (symptom onset <65 years); (2) have mild or moderate cognitive impairment, as measured by a Mini-Mental State Examination [39,40] score between 22–28); (3) able to tolerate approximately 100-min online sessions; (4) able to understand and communicate in English; (5) have functional hearing (with or without hearing aids or other devices); (6) have access to the internet and a device (laptop or tablet).

#### 2.3.2. Procedure and Randomisation

Interested and potential dyads will receive a phone call whereby the above criteria will be discussed to check eligibility for entry. Following the screening, informed consent will be obtained by caregivers and people living with YOD. Dyads will be randomly allocated a computer-generated number 0 or 1. If they receive a “1”, they will be randomised into the intervention MAPS group. Each intervention group will have a maximum of seven dyads. For those dyads who receive a “0”, they will be allocated to the waitlist control group and be invited to participate in MAPS following the completion of the intervention. All eligible participants will be administered the outcomes measures (see below).

### 2.4. Measures

Demographic information (age, gender), information about caregiving (duration and approximate hours per day), MMSE scores, treatments, and dementia type if known and severity (mild, moderate, severe pre- and post- program) will be collected. To detect changes in anxiety and depression in caregivers (primary outcome), the DASS-21 will be administered pre- and post-program. The DASS is a 21-item scale composed of three sub-scales with 7 items each, measuring depressive, anxiety, and stress symptoms, and has good psychometric properties (reliability and validity). Participants are asked to rate each item based on how they have been feeling the past week with a 0 (not at all) to 3 (most of the time) scale for responses. The score is multiplied by 2 to make scores equivalent to the DASS-42 [38].

Secondary outcome measures for the caregivers collected pre- and post-MAPS are:

Coping: The Brief COPE [41] is a 28-item questionnaire measuring three different types of coping, emotion-, problem- and avoidant-coping mechanisms, and has strong psychometric properties.

Revised Scale for Caregiving Self-Efficacy: [42] is a 15-item scale with 3 subscales that measure self-efficacy. It captures caregiver perspectives on respite, responding to disruptive behaviours, and controlling upsetting thoughts about caregiving. It has strong psychometric properties.

Social Connectedness: The Friendship Scale [43] is a 6-item scale measuring 6 dimensions that contribute to social isolation and social connection of older adults. Based on the last 4 weeks, participants rate each item on a 5-point Likert scale (0 = Almost always to 4 = Not at all) with total scores ranging from 0–24. Psychometric testing suggests good reliability (Cronbach’s α = 0.83) and sensitivity to change for known dimensions impacting social isolation.

Secondary outcome measures for the people living with YOD collected pre- and post-MAPS are:

Social Connectedness: The Friendship scale [43], as described above.

Mental health: The DASS-21 [38], as described above.

Behaviours using the Neuropsychiatric Inventory–brief form (NPI-Q) [44]. This brief version of the NPI-NH has 12 items measuring behaviours such as depression, hallucinations, and agitation/aggression. Caregivers are asked to rate the severity of how it affects the people with YOD (Mild, moderate, severe) and the distress the caregiver experiences (0 nil to 5 extreme distressing).

Depression, using the Cornell Scale for Depression in Dementia, CSDD [45]. This 19-item scale asks the caregivers to rate the people with dementia’s mood and behaviour, rated from 0 (absent) to 2 (severe). Scores > 10 indicate probable major depression and scores > 18 indicate definite major depression.

A process evaluation will be undertaken in order to gain a better understanding of participants’ experiences and the mechanisms of change. Focus group questions will explore participants’ reasons for joining and experiences of the program including any positive or negative aspects of the program, and what changes (if any) they noticed in the group, themselves, their partner, or their use of music. Questions will also focus on any perceived impact of the program on their relationships and feelings of connectedness (both between group members and within the dyadic relationship). The focus group will also include discussion around the continuation of the private Facebook group and if this is something participants wish to continue following completion of the research.

### 2.5. Data Collection and Management and Statistical Analysis Plan

All self-report measures will be collected online using REDCaP and stored electronically. Dyads successfully screened into the program will be contacted and invited to an online session where the pre-program measures will be completed. Post-MAPS quantitative outcome measures will be collected by members of the research team who are masked to allocation. Post-MAPS qualitative evaluation will be audio-recorded and transcribed for analysis.

Our primary outcome measure is an improvement in the DASS-21 (pre- and post-), with a 4-point reduction (SD of differences = 8). Using β = 80% (power) and α = 0.05 and medium effect size = 0.5 (*p* < 0.05), we will require a sample size of n = 54 dyads. Accounting for 10% attrition, we aim to recruit 60 dyads. We thus anticipate approximately 9 “waves” of MAPS (i.e., 7 dyads per wave, accounting for at least one drop-out per wave). As the program involves sequential weekly sessions, we will not be able to replace drop-outs during the program. For demographic analysis, we will use frequencies and descriptive statistics. Categorical tests such as χ^2^ will be used to compare groups on factors such as dementia types and gender. Paired *t*-tests will be used to compare pre- and post- primary and secondary outcomes and we will also calculate effect sizes to determine the size of the changes in DASS-21 and other measures. Statistical Program for Social Sciences (SPSS) version 27 (IBM) will be used for quantitative analyses. Qualitative analysis of focus group interviews will use Interpretative Phenomenological Analysis to synthesise their experiences, benefits, and limitations of participating in the project.

### 2.6. Public and Patient Involvement (PPI)

The design of this project emerged from engagement with people with lived experience. From the beginning of this project, we have engaged closely with caregivers of people living with YOD who have been involved with the program development in terms of conception, the conceptual framework, session duration and frequency, and content as well as providing guidance on implementation. Feedback confirmed that the session content was appropriate, including a balance between “being positive and talking about living well with dementia but also addressing the challenges”. They reported that many programs only ever focused on shifts towards being positive without a full acknowledgement of the frustrations and challenges that are very real and impactful. Several changes were made to our initial planned program including increasing session lengths to 100-min (because we had underestimated the amount of time participants will want to share and talk), reordering some of the topics (e.g., offering information about support services in session 1 rather than session 3), and creating a manual or workbook to be sent to participants in advance of the program. Caregivers have reviewed the content of the workbooks, which will be provided to the dyads and we have ensured that their suggestions have been incorporated.

During the project, the PPI group will be consulted on all matters concerning implementation, consulting on recruitment approaches, and the development and design of the Facebook social networking component. They will also be involved in reviewing the revised program once the current pilot study has been completed and recommendations integrated into the final program design. The PPI will also have an opportunity to have input into the dissemination plan and how the songs created by participants might be shared and used to advocate for better services for people with YOD and their caregivers.

## 3. Discussion

This paper presents a protocol for a novel intervention combining therapeutic songwriting, psychological therapy, and social media, administered through an online platform for people living with YOD and their caregivers (i.e., dyad-based). Being manualised means that the program can easily be replicable and administered. Being online will enable flexibility, provide the ability to reach dyads who live in rural and regional communities as well considers the challenges associated with a COVID world. A closed social networking group will provide a platform for psychoeducation, sharing experiences, providing support, and providing opportunities for connection and interactions.

Our protocol has some limitations. The intervention is not targeted to subtypes of YOD and hence the sessions are not specific to challenges associated with specific dementia types such as Alzheimer’s disease or frontotemporal dementia, but more about the psychosocial challenges faced within the dyad. However, due to heterogeneity of presentation in YOD with difficulties in obtaining a diagnosis, we feel that our intervention addresses the lack of interventions in this younger-onset space for people with dementia and their caregivers. Similarly, we only target those with mild/moderate severity so this intervention does not consider those who are in the advanced stages of dementia, who would have different dyadic and caregiving needs. In addition, as a dyadic intervention, this framework does not include the wider family or non-spousal caregivers.

Living with and supporting a person living with YOD comes with unique challenges. Currently, there are very few psychosocial interventions aimed specifically at the dyad. Interventions need to be appropriate for the younger age group, be person-centred and address the relevant psychosocial issues. We have described a novel program involving both the caregivers and people living with YOD, using combined evidence-based strategies of therapeutic songwriting and psychological techniques administered online, with trained facilitators, aiming to improve mental health and social connections. If the program is successful, we anticipate that the program will be easily scalable and able to benefit more younger-onset dementia dyads.

## Figures and Tables

**Figure 1 brainsci-12-00503-f001:**
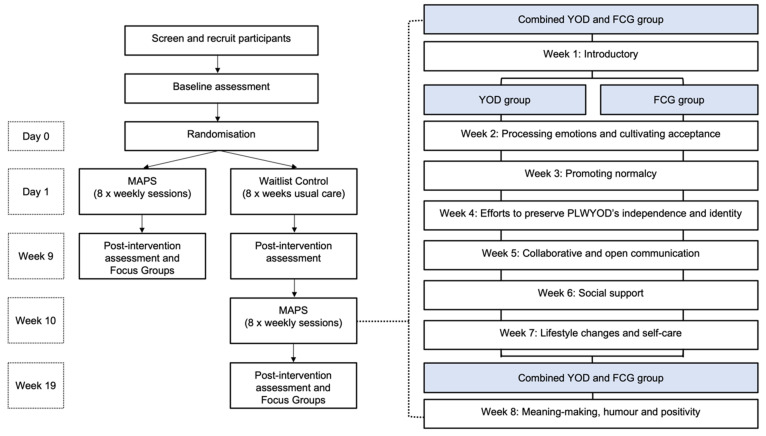
Structure of the program (FCG family caregivers; MAPS Music And Psychology & Social connections; YOD younger-onset dementia).

**Figure 2 brainsci-12-00503-f002:**
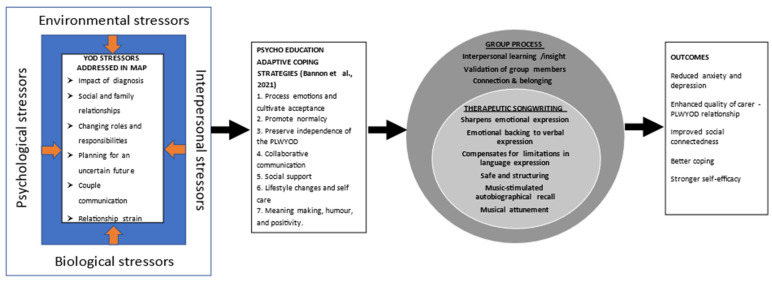
The theoretical framework, based on Bannon et al. (2021) (PLWYOD people living with younger-onset dementia).

## Data Availability

The data is not available as the trial has not commenced. Data may be made available depending upon reasonable request to the authors.

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
