# Peer review of "Music and Psychology & Social Connections Program: Protocol for a Novel Intervention for Dyads Affected by Younger-Onset Dementia"

_brainsci, 2022, doi:10.3390/brainsci12040503_

Round 1
Reviewer 1 Report
Thank you for the opportunity to review this paper. It is a quite interesting work and also of great significance regarding nonpharmacological interventions in this population. However, I would like to address the comments below:
Abstract:
- Authors say «MAPS has the potential for scalability with benefits for those affected by younger-onset 30 dementia.» This comment is not concluded by the content of the manuscript.
- Abstract, in general, is not well written. Method is missing.. It seems that authors wrote only the Intro and the Conclusion (however the effectiveness of MAP is not provided in the current manuscript)
Introduction
- Authors claim that they want to implement this intervention in Young onset dementia. Which kind?? According to the already existing knowledge, YOD symptomatology differs across the different underlying pathologies. It would be maybe better to recruit e.g patients with early Alzheimer’s or at least mention patients’ diagnosis begore starting the intervention.
- Authors refer to Bannon et al’s (2021) adaptive coping framework. Could it be explained in more detail?
- Are there any longitudinal studies concerning dyad interventions?
- Lines 47-54 : which adverse events happen to caregivers and which refer to patients…it is not clear
- Lines 93-97: Although this intervention was arguably more comprehensive compared to ANSWERS……which might be perceived as less overwhelming. Authors state that Danish study can be assumed as more effective, because shorter programs are preferable. Do you think that this is the reason between studies’ discrepancy?Do they think that longer interventions are less effective?
- Are authors aware of previous studies measuring a) depressive symptoms in people living with YOD; b) the effectiveness and types of coping strategies utilised by caregivers; c) levels of social connectedness in people living with YOD and their caregivers; and d) the levels of neuropsychiatric symptoms in the people living with YOD?
Materials and Methods
- Authors are encouraged to provide a better and more structured description of the program. Figure 2 should be accompanied with a relevant text.
- Please provide some more information about if patients will have undergone any other kind of treatment (especially non pharmacological ones), Additionally, disease’s characteristics could intervene in the study’s final outcome. In which way authors will manage this issue?
- Lines 221-224 “The final session (week 8) will bring together the dyads to explore meaning-making, humour and positivity and provide an opportunity to share the group composed songs for closure. The songs produced by MAPS will be audio-recorded so that these can be played during presentations to demonstrate what can be achieved.” Can you please describe it more detail?In which way humor and positivity will be introduced in the program?
- In which was authors will manage any Drop outs??
Discussion
- Authors are encouraged to write down by which way their intervention will improve the already existing interventions in YOD population. Which studies already exist and which is the complementary point they would like to address?
Please provide the Limitations in the end of the manuscript.
Author Response
Thank you Reviewer for reading our manuscript and providing these detailed comments and suggestions for improvement!
Abstract:
- Authors say «MAPS has the potential for scalability with benefits for those affected by younger-onset 30 dementia.» This comment is not concluded by the content of the manuscript.
We apologise for this omission and have removed this from the abstract (pg. 1).
- Abstract, in general, is not well written. Method is missing.. It seems that authors wrote only the Intro and the Conclusion (however the effectiveness of MAP is not provided in the current manuscript)
Thank you for your critique. We have amended the abstract for clarity and also to highlight the Methods which reports on the number of sessions in the intervention and what they entailed. As this is a protocol paper we have not been able to describe the effectiveness of MAPS yet so did not put this in the Abstract nor manuscript.
Introduction
- Authors claim that they want to implement this intervention in Young onset dementia. Which kind?? According to the already existing knowledge, YOD symptomatology differs across the different underlying pathologies. It would be maybe better to recruit e.g patients with early Alzheimer’s or at least mention patients’ diagnosis begore starting the intervention.
Because younger-onset dementia has different symptoms and heterogeneous causes, we have not specified which type of dementia and are not restricting the type of dementia we are targeting for our intervention. But certainly at recruitment level we will be asking the dyads what is the type/cause of younger-onset dementia. We add on pg. 7 “there is no restriction on the type of YOD” and also that we will be collect information on the type of dementia and any medications/treatments” (pg. 7). We also add that this is a limitation of our intervention (pg.9)
- Authors refer to Bannon et al’s (2021) adaptive coping framework. Could it be explained in more detail?
We expand on Bannon et al.’s framework on pg. 5. under 2.2.1 Conceptual Framework.
- Are there any longitudinal studies concerning dyad interventions?
We were unable to find any longitudinal studies of dyad interventions for younger-onset dementia.
- Lines 47-54 : which adverse events happen to caregivers and which refer to patients…it is not clear
- Lines 93-97: Although this intervention was arguably more comprehensive compared to ANSWERS……which might be perceived as less overwhelming. Authors state that Danish study can be assumed as more effective, because shorter programs are preferable. Do you think that this is the reason between studies’ discrepancy?Do they think that longer interventions are less effective?
We apologise for the lack of clarity in this paragraph! We have simplified this to “it is unclear whether length of intervention and which elements of an intervention (for example, group vs individual, psychoeducation vs counselling) are more effective in improving depression and anxiety” (pg. 2/3)
- Are authors aware of previous studies measuring a) depressive symptoms in people living with YOD; b) the effectiveness and types of coping strategies utilised by caregivers; c) levels of social connectedness in people living with YOD and their caregivers; and d) the levels of neuropsychiatric symptoms in the people living with YOD?
As far as we are aware from the literature, there are few intervention studies in people with younger-onset dementia. There are 2 relatively recent reviews on this (Richardson et al. 2016 and Aplaon et al. 2017) with four studies with sample sizes ranging from 1 – 9. These occurred in the workplace where the person with younger-onset dementia had a “buddy” who helped to facilitate ongoing work and improved social connectedness. There is literature reporting on neuropsychiatric symptoms, including depressive symptoms, in people with younger-onset dementia. The NEED-YD study (the Dutch group) and BEYOND study (people with younger-onset dementia in aged care facilities) and a recent Australian study compared neuropsychiatric symptoms in people with older- and younger-onset dementia (Loi et al. 2022). Because we wished to focus our literature search on dyads, we did not discuss these studies which included only individual people with younger-onset dementia. However, in the Discussion (pg. 9), we acknowledge that our intervention fills a gap in lack of interventions in this younger-onset dementia space.
The two studies which focused on caregivers of people with younger-onset dementia (RHAPSODY and Partner in Balance) are described in the manuscript.
Materials and Methods
- Authors are encouraged to provide a better and more structured description of the program. Figure 2 should be accompanied with a relevant text.
We agree and have made numerous amendments in the Methods as well as more information for Figure 2 (including a title for Figure 2).
- Please provide some more information about if patients will have undergone any other kind of treatment (especially non pharmacological ones), Additionally, disease’s characteristics could intervene in the study’s final outcome. In which way authors will manage this issue?
Both excellent points which we have amended the Methods to improve our manuscript. On pg. 7, under 2.4 Measures, we have added that we will collect information regarding medications and other treatments. We also have added that we will ask caregivers pre- and post-program, the severity of the dementia (mild, moderate and severe).
- Lines 221-224 “The final session (week 8) will bring together the dyads to explore meaning-making, humour and positivity and provide an opportunity to share the group composed songs for closure. The songs produced by MAPS will be audio-recorded so that these can be played during presentations to demonstrate what can be achieved.” Can you please describe it more detail?In which way humor and positivity will be introduced in the program?
The final session is intended to act as a reflection and tying up of the whole program with the whole group (caregivers and people living with younger-onset dementia). We want to facilitate and discuss the previous sessions, including the positive aspects and using humour as one reframing technique to assist this. Playing the songs back to the group would also demonstrate what the group has achieved tangibly as well provide a positive reminder of the program (p.6, 2.2.2 Description of the MAPS program).
- In which was authors will manage any Drop outs??
On pg. 8 (2.5 Data collection and management and statistical analysis plan), we consider drop-outs by reporting that they will not be replaced as the program involves weekly sessions where content is continually built upon.
Discussion
- Authors are encouraged to write down by which way their intervention will improve the already existing interventions in YOD population. Which studies already exist and which is the complementary point they would like to address?
This is an excellent recommendation for our Discussion (pg. 9) and we have added that that our intervention fills a gap in the lack of interventions for those affected by younger-onset dementia.
Please provide the Limitations in the end of the manuscript.
We have added all of these in our Discussion section pg. 9. Specifically that we are not targeting a specific type of younger-onset dementia and that the intervention is only for mild/moderate severity and thus does not consider those with more advanced stages of dementia. In addition, being dyad-based, the framework does not include the wider family as caregivers, nor non-spousal caregivers.
Reviewer 2 Report
Dear author,
The article shows the lack of psychosocial interventions addressing the specific needs of people with dementia before the age of 65. The article presents a novel intervention that combines evidence-based therapy, therapeutic song composition, cognitive behavioural therapy. The aim was to assess whether this intervention with these characteristics improves depressive symptoms, anxiety, and stress in the caregivers of these older people and also to improve the older people's depressive symptoms and coping skills. The intervention is based on an 8-week programme.
First of all, congratulations on the work done and the effort involved in conducting research of this nature.
It provides good conceptualisation, design, good analysis and discussion.
Congratulations on the work done.
However, I would like to suggest a number of recommendations:
Expand the introduction and, above all, the discussion with articles for debate.
Incorporate if you wish these three reference quotes from 2021 -2020 in your text, on the importance of socio-contextual variables that influence and are not forgotten in the context of the older person:
1) The practice of vigorous physical activity is related to a higher educational level and income in older women Zapata-Lamana, R., Poblete-Valderrama, F., Cigarroa, I., Parra-Rizo, M.A. International Journal of Environmental Research and Public Healththis link is disabled, 2021, 18(20), 10815 doi: 10.3390/ijerph182010815 ""We have to take into account socio-contextual factors that affect the health of older people, in the case of older women, higher levels in some of the socio-environmental aspects of quality of life are associated with better health care, such as higher levels of education and income in a better quality of life, so social inequalities may interfere with lifestyles".
2) Parra-Rizo, M.A. & Sanchís-Soler, G. (2021). Physical activity and the Improvement of autonomy, functional ability, subjective health, and social relationships in women over the age of 60. International Journal of Environmental Research and Public Health, 18(13):6926, doi: 10.3390/ijerph18136926 "A socio-contextual factor of the environment that is included in the improvement of quality of life in older people is the practice of physical activity. In fact, those with high levels of physical activity obtained better levels of functional capacity and autonomy. In addition, dissatisfaction with one's own health is associated with low levels of physical activity. The practice of light physical exercise in older women promotes greater autonomy and functional capacity for activities of daily living, which results in independence in daily life as well as fostering social bonds, as well as obtaining greater satisfaction with their own health, with the socioemotional benefits that this can bring. (Parra-Rizo & Sanchís-Soler, 2021).""
3) Parra-Rizo, M.A. & Sanchís-Soler, G. (2020). Satisfaction with life, subjective well-being and functional skills in active older adults based on their level of physical activity practice. International Journal of Environmental Research and Public Health, 18;17(4):1299, doi: 10.3390 / ijerph17041299 "Older adults with a high level of physical activity have been found to have greater life satisfaction, subjective well-being, better functional skills and fewer difficulties in performing activities of daily living, as well as greater self-esteem and health (Parra-Rizo & Sanchís-Soler, 2020)".
In the discussion, in addition, to make it much more complete:
-Incorporate what theoretical implications this work has for the scientific community. What does this work imply for researchers.
-Incorporate what practical implications this work has for the community/society/population.
-Incorporate what implications this work has on the social level.
-Incorporate limitations of their work.
-Incorporate strengths of your work.
-Incorporate future lines of research.
Best regards,
Author Response
Thank you Reviewer for your taking the time to provide us with your suggestions for improvement.
Expand the introduction and, above all, the discussion with articles for debate.
Incorporate if you wish these three reference quotes from 2021 -2020 in your text, on the importance of socio-contextual variables that influence and are not forgotten in the context of the older person.
We thank the Reviewer very much for these comments and also for the three references which the titles suggest that they report on older adults and the benefits of physical activity. We agree that the sociocultural context is also important for people living with younger-onset dementia and their caregivers and thus this protocol focuses on social connections, but it was beyond the scope of the study to examine physical activity .
In the discussion, in addition, to make it much more complete:
-Incorporate what theoretical implications this work has for the scientific community. What does this work imply for researchers.
-Incorporate what practical implications this work has for the community/society/population.
-Incorporate what implications this work has on the social level.
-Incorporate limitations of their work.
-Incorporate strengths of your work.
-Incorporate future lines of research.
We appreciate these suggestions and have added these in our Discussion section pg.9.
Limitations: Specifically that we are not targeting a specific type of younger-onset dementia and that the intervention is only for mild/moderate severity and thus does not consider those with more advanced stages of dementia. In addition, being dyad-based, the framework does not include the wider family as caregivers, nor non-spousal caregivers. Thus future lines of research would include developing interventions to suit these other groups.
Strengths: The manualised nature of this intervention means that it can be easily replicable.
Practicalities: Being online, the intervention is flexible to the needs of this younger group and can reach those who live in regional and rural areas who might otherwise miss out on these programs.
It should be noted, however, that this is a protocol paper, and as such the implications of the work will become clearer when we have results to report.

Round 2
Reviewer 2 Report
Dear author,
Congratulations for the substantial improvement of your article, in this way, you will increase your visibility, that readers will cite you and have their interest in reading this improved article. congratulations for the work and effort that this entails.
Best regards